# Bias and Generalization in Deep Generative Models: An Empirical Study

**Shengjia Zhao**[†]**, Hongyu Ren**[†]**, Arianna Yuan, Jiaming Song, Noah Goodman, Stefano Ermon**
Stanford University
{sjzhao,hyren,xfyuan,tsong,ngoodman,ermon}@stanford.edu

## Abstract

In high dimensional settings, density estimation algorithms rely crucially on their inductive bias. Despite recent empirical success, the inductive bias of deep generative models is not well understood. In this paper we propose a framework to systematically investigate bias and generalization in deep generative models of images. Inspired by experimental methods from cognitive psychology, we probe each learning algorithm with carefully designed training datasets to characterize when and how existing models generate novel attributes and their combinations. We identify similarities to human psychology and verify that these patterns are consistent across commonly used models and architectures.

## 1 Introduction

The goal of a density estimation algorithm is to learn a distribution from training data (Figure 1,A). However, unbiased and consistent density estimation is known to be impossible [1, 2]. Even in discrete settings, the number of possible distributions scales *doubly* exponentially w.r.t. dimensionality [3], suggesting extremely high data requirements. As a result, the assumptions made by a learning algorithm, or its inductive bias, are key when practical data regimes are concerned. For simple density estimation algorithms, such as fitting a Gaussian distribution via maximum likelihood, we can easily characterize the distribution that is produced given some training data. However, for complex algorithms involving deep generative models such as Generative Adversarial Networks (GAN) and variational autoencoders (VAE) [4–8], the nature of the inductive bias is very difficult to characterize.

In the absence of insights in analytic form, a possible strategy to evaluate this bias is to probe the input-output behavior of the learning algorithm. The challenge with this approach is that both inputs and outputs are high dimensional (e.g., distributions over images), making it difficult to exhaustively characterize the input-output relationship. A strategy for studying high-dimensional objects is to project them onto a lower dimensional space where analysis is feasible. In fact, similar problems have long challenged cognitive psychologists. As visual cognitive functions are extremely complex, cognitive psychologists and neuroscientists have developed controlled experiments to investigate the visual system. For example, experiments on perception and representation of shape, color, numerosity, etc., have led to important discoveries such as ensemble representation [9], prototype enhancement effect [10], and Weber's law [11].

We propose to adopt experimental methods from cognitive psychology to characterize the generalization biases of machine intelligence. To characterize the input-output relationship of an algorithm, we explore its behavior by projecting the image space onto a carefully chosen low dimensional feature space. We select several features that are known to be important to humans, such as shape, color, size, numerosity, etc. We systematically explore these dimensions by crafting suitable training datasets and measuring corresponding properties of the learned distribution. For example, we ask, after training on a dataset with red and yellow spheres, and red cubes, will the model generates yellow cubes, as a result of its inductive bias?

---

[†] co-first authorship.

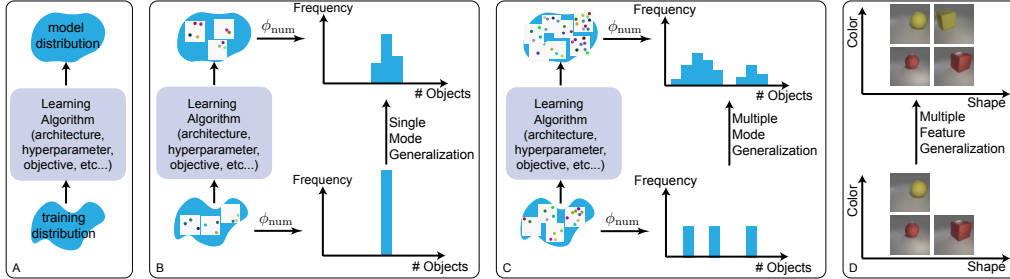

Figure 1: **A:** A deep generative model can be thought of as a black box that we probe with carefully designed training data. **B:** First we examine the learned distribution when training data takes a single value for some feature (e.g., all training images have 3 objects). **C:** Next we study input distributions with multiple modes for some feature (e.g., all training images have 2, 4 or 10 objects). **D:** We explore the behavior of the model when the training data has multiple modes for multiple features.

Using this framework, we are able to systematically evaluate generalization patterns of state-of-the-art models such as GAN [5] and VAE [4]. Surprisingly, we found these patterns to be consistent across datasets, models, and hyper-parameters choices. In addition, some of these patterns have striking similarities with previously reported experiments in cognitive psychology. For example, when presented with a training set where all images contain exactly 3 objects, both GANs and VAEs typically generate 2-5 objects, with a log-normal shaped distribution (Figure 1,B). If the training set contains multiple modes (e.g., all images contain either 2 or 10 objects) we observe a behavior similar to that of a linear filter — the algorithm acts as if it is trained separately on 2 and 10 objects and then averages the two distributions. An exception is when the modes are close to each other (e.g., 2 and 4 objects) where we observe prototype enhancement effect [10]: the learned distribution assigns higher probability to the mean number of objects (3 in our example), even though no image with 3 objects was present in the training set (Figure 1,C). Finally for multiple features, when the training set only contains certain combinations, e.g., red cubes but not yellow cubes (Figure 1,D), we find that the learning algorithms will memorize the combinations in the training set when it contains a small number of them (e.g., 20), and will generate new combinations (not in the training set) when there is more variety (e.g., 80). We study the amount of novel combinations generated as a function of the number of combinations in the training set. For all of the observations we find consistent results across a diverse set of tasks (CLEVR, colored dots, MNIST), training objectives (GAN or VAE), architectures (convolutional or fully connected layers), and hyper-parameters.

## 2 Density Estimation, Bias, and Generalization

Let $\mathcal{X}$ be the input space (e.g., images), and let $\mathcal{D} = \{\mathbf{x}_1, \cdots, \mathbf{x}_n\}$ be a training dataset sampled i.i.d. from an underlying distribution $p(\mathbf{x})$ on $\mathcal{X}$. The goal of a density estimation algorithm $\mathcal{A}$ is to take $\mathcal{D}$ as input and produce a distribution $q(\mathbf{x})$ over $\mathcal{X}$ that is "close" to $p(\mathbf{x})$. Crucially, the same algorithm $\mathcal{A}$ should work (well) on a range of input datasets, sampled from different distributions $p(\mathbf{x})$.

However, estimating $p(\mathbf{x})$ is difficult [1]. In fact, even the simplified task of estimating the support of $p(\mathbf{x})$ is challenging in high dimensions. For example, natural images vary along a large number of axes, e.g., the number of objects present, their type, color, shape, position, etc. Because the number of possible combinations of these attributes or features grows exponentially (w.r.t the number of possible features), the size of $\mathcal{D}$ is often exponentially small compared to the support of $p(\mathbf{x})$ in this feature space. Therefore strong prior assumptions must be made to generalize from this very small $\mathcal{D}$ to the *exponentially larger* support set of $p(\mathbf{x})$. We will refer to the process of producing $q(\mathbf{x})$ from $\mathcal{D}$ as **generalization**, and any assumptions used when producing $q(\mathbf{x})$ from $\mathcal{D}$ as **inductive bias** [12].

Deep generative modeling algorithms implicitly use many types of inductive biases. For example, they often involve models parameterized with (convolutional) neural networks, trained using adversarial or variational methods. In addition, the training objective is typically optimized by stochastic gradient descent, contributing to the inductive bias [13, 14]. The resulting effect of these combined factors is difficult to study theoretically. As a result, empirical analysis has become the primary approach. For example, it has been shown that on multiple natural image datasets, the learned distribution produces novel examples that generalize in meaningful ways, going beyond pixel space interpolation [15–17]. However these studies are not systematic. They do not answer questions such as how the learning

algorithm will generalize given a new dataset, or provide insight into exactly which inductive biases are involved. The lack of systematic study is due to the high dimensionality of both the input dataset $\mathcal{D}$ and the output distribution $q(\mathbf{z})$. In fact, even evaluating how "close" the learned distribution $q$ is to $p$ is an open question, and there is no commonly accepted evaluation metric [18–20]. Therefore, to examine the inductive bias we need to design settings where the training and output distributions can be exactly characterized and compared.

## 3 Exploring Generalization Through Probing Features

We take inspiration from cognitive psychology, and provide a novel framework to analyze empirically the inductive bias of generative algorithms via a set of probing features. We focus on images but the techniques can also be applied to other domains.

Let $\mathcal{S} \subset \mathcal{X}$ be the support set of $p(\mathbf{x})$. We define a set of (probing) features as a tuple of functions $\boldsymbol{\phi} = (\phi_1, \cdots, \phi_k)$ where each $\phi_i$ maps an input image in $\mathcal{S}$ to a value. For example, one of the features $\phi_i : \mathcal{S} \to \mathbb{N}$ may map the input image to the number of objects (numerosity) in that image (Figure 1BC). We denote the range of $\boldsymbol{\phi}$ as the feature space $\mathcal{Z}$. For any choice of $p(\mathbf{x})$, with a slight abuse of notation, we denote $p(\mathbf{z})$ as the (induced) distribution on $\mathcal{Z}$ by $\boldsymbol{\phi}(\mathbf{x})$ when $\mathbf{x} \sim p(\mathbf{x})$. Intuitively $p(\mathbf{z})$ is the projection of $p(\mathbf{x})$ onto the feature space $\mathcal{Z}$.

When a learning algorithm $\mathcal{A}$ produces a learned distribution $q(\mathbf{x})$, we also project it to feature space using $\boldsymbol{\phi}$. Our goal is to investigate how $p(\mathbf{z})$ differs from $q(\mathbf{z})$, i.e. the generalization behavior of the learning algorithm restricted to the feature space $\mathcal{Z}$. In the input space $\mathcal{X}$ even evaluating the distance between $p(\mathbf{x})$ and $q(\mathbf{x})$ is difficult, while in feature space $\mathcal{Z}$ we can not only decide if $q(\mathbf{z})$ is different from $p(\mathbf{z})$ but also characterize *how they are different*. For example, if $p(\mathbf{z})$ is a distribution over images with red and blue triangles (▲,▲) and red circles (●), we can investigate whether $q(\mathbf{z})$ generalizes to blue circles (●). We can also investigate the number of colors for circles that must be in the training data so that $q(\mathbf{z})$ generates circles of all colors. Such questions are important to characterize the inductive bias of existing generative modeling algorithms.

Related ideas [21] have been previously used to evaluate the distance between $p(\mathbf{x})$ and $q(\mathbf{x})$. In particular, the FID score [22], the mode score [23] and the Inception score [24] use hidden features/labels of a pretrained CNN classifier as $\boldsymbol{\phi}$, and measure the performance of generative modeling algorithms by comparing $p(\mathbf{z})$ and $q(\mathbf{z})$ under this projection. In contrast, because we want to study the exact difference between $p(\mathbf{z})$ and $q(\mathbf{z})$, we choose $\boldsymbol{\phi}$ to be interpretable high level features inspired by experimental work in cognitive psychology, e.g. numerosity, color, etc.

Using low dimensional projection function $\boldsymbol{\phi}$ has an additional benefit. Because $\mathcal{Z}$ is low dimensional and discrete in our synthetic datasets, we are essentially in the infinite data regime. In all of our experiments, the support of $p(\mathbf{z})$ does not exceed 500, so we accurately approximate $p(\mathbf{z})$ [25] with a reasonably sized dataset (100k-1M examples in our experiments). The interesting observation is that even though $\mathcal{D}$ is a very accurate approximation of $p(\mathbf{z})$, the learned distribution $q(\mathbf{z})$ is not, so this simplified setting is sufficient to reveal many interesting inductive biases of the modeling algorithms.

**Feature Selection and Evaluation**   We select features $\boldsymbol{\phi}$ that satisfy two requirements: 1) they are important to human perception and have been studied in cognitive psychology, and 2) they are easy to evaluate either by reliable algorithms or human judgment. The features studied include numerosity, shape, color, size, and location of each object. For numerosity and shape we use independent evaluations by three human evaluators. The other features are easy to evaluate by automated algorithms. More details about evaluation are presented in the appendix.

**Models**   To ensure that the result is not sensitive to the choice of model architecture and hyper-parameters, we use two very different model families: GAN (WGAN-GP [26]) and VAE [4]. We also use different network architectures and hyper-parameter choices, including both convolutional networks and fully connected networks. We will present the experimental results for WGAN-GP with convolutional networks in the main body, and results for other architectures in the appendix. Surprisingly, we find *fairly consistent results for these very different models and objectives*. Whenever they differ, we will explicitly mention the differences in the main body.

## 4 Characterizing Generalization on an Individual Feature

In this section we explore generalization when we project the input space $\mathcal{X}$ to a *single* feature (i.e., $p(\mathbf{z})$ is a one-dimensional distribution). We first analyze the learning algorithm's output $q(\mathbf{z})$ when

the feature we manipulate contains only one value, i.e., $p(\mathbf{z})$ is a delta function/unit impulse. We ask questions such as: when all images in the training set depict five objects, how many objects will the generative model produce? One might expect that since the feature takes a single value, and we have hundreds of thousands of distinct examples, the learning algorithm would capture exactly this fixed feature value. However this is not true, indicating strong inductive bias.

We call the learned distribution $q(\mathbf{z})$ when the input distribution has a single mode the **impulse response** of the modeling algorithm. We borrow this terminology from signal processing theory because we find the behavior similar to that of a linear filter: if $p(\mathbf{z})$ is supported on multiple values, the model's output $q(\mathbf{z})$ is a **convolution** between $p(\mathbf{z})$ and the model's impulse response. An exception is when two modes of $p(\mathbf{z})$ are close together. In this case we find **prototype enhancement effect** and the learning algorithm produces a distribution that "combines" the two modes. Finally we justify our approach of studying each single features individually by showing that the learning algorithm's behavior on each feature is mostly **independent** of other features we study.

## 4.1 Generalization to a Single Mode

### 4.1.1 Numerosity

**Experimental Settings**   We use two different datasets for this experiment: a toy dataset where there are $k$ non-overlapping dots (with random color and location) in the image, as in the numerosity estimation task in cognitive psychology [27, 28], and the CLEVR dataset where there are $k$ objects (with random shape, color, location and size) in the scene [29]. More details about the datasets are provided in the Appendix. Example training and generated images are shown in Figure 2, left and right respectively.[1]

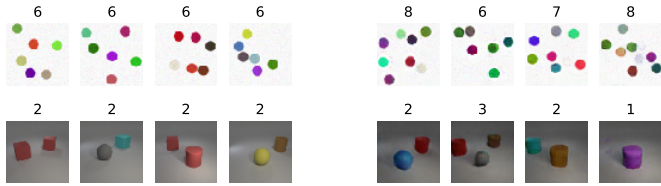

Figure 2: Example training (left) and generated images (right) with annotated numerosity. In this example, all training examples contain six dots (top) or two CLEVR objects (bottom), while the generated examples on the right often contain a different number of dots/objects.

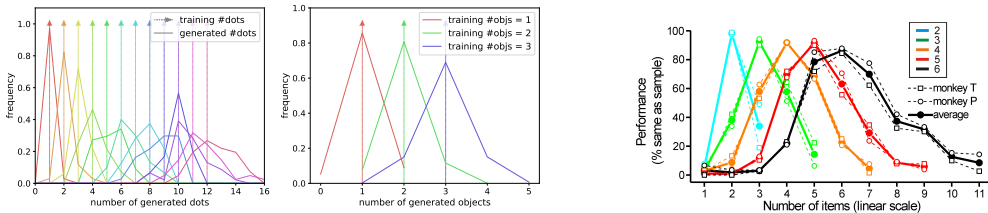

Figure 3: **Left:** Distribution over number of dots. The arrows are the number of dots the learning algorithm is trained on, and the solid line is the distribution over the number of dots the model generates. **Middle:** Distribution over number of CLEVR objects the model generates. Generating CLEVR is harder so we explore small numerosities, but the generalization pattern is similar to dots. **Right:** Numerosity perception of monkeys [30]. Each solid line plots the likelihood a monkey judges a stimuli to have the same numerosity as a reference stimuli. Figure adapted from [30]

**Results and Discussion**   As shown in Figure 2 and quantitatively evaluated in Figure 3, in both colored dots and CLEVR experiments, *the learned distribution does not produce the same number of objects as in the dataset on which it was trained*. The distribution of the numerosity in the generated images is centered at the numerosity from the dataset, with a slight bias towards over-estimation. For example, when trained on images with six dots (Figure 2, cyan curve) the generated images contain anywhere from four to nine dots.

Researchers have found neurons that respond to numerosity in human and primate brains [27, 28]. From both behavioral data and neural data, two salient properties about these neurons were

---

documented [31]: 1) larger variance for larger numerosity, and 2) asymmetric response with more moderate slopes for larger numerosities compared to smaller ones [31, 30] (Figure 3 right). It is remarkable that deep generative models generalize in similar ways w.r.t the numerosity feature.

### 4.1.2 Color Proportion

**Experimental Settings**    For this feature we use the dataset shown in Figure 4. Each pie has several properties: proportion of red color $\mathbf{z}_{red}$, size $\mathbf{z}_{size}$, and location $\mathbf{z}_{loc}$. In these experiments we choose the proportion of red to be 10%, 30%, 50%, 70%, 90% respectively, while the other features (size and location) are selected uniformly at random within the maximum range allowed in the dataset. Details about the dataset can be found in the Appendix.

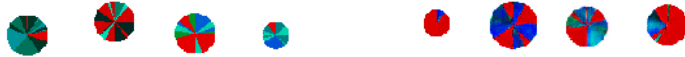

Figure 4: Example images from the training set (Left) and generated by our model (Right). Each circle has a few properties: proportion of the pie that takes color red, radius of the pie and location (of center of circle) on the x and y axis.

**Results and Discussion**    The results are shown in Figure 5(Left). We find that the learned feature distribution $q(\mathbf{z})$ is well approximated by a Gaussian centered at value the model is trained on. What is notable is that the Gaussian is sharper for small (10%) and large proportions (90%). This is consistent with Weber's law [11], which states that humans are in fact sensitive to relative change (ratio) rather than absolute change (difference) (e.g., the difference between 10% to 12% is more salient compared to 40% to 42%). Unlike numerosity, generalization in this domain is symmetric.

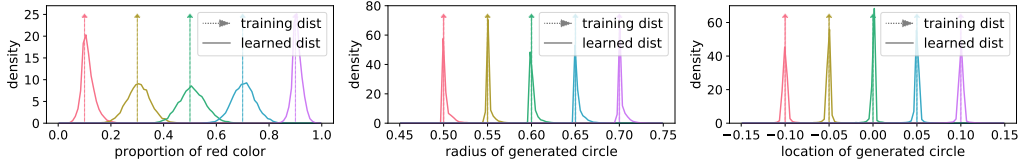

Figure 5: Generated samples for each feature (proportion of color (**left**), size (**middle**) and location **right**)). The dashed arrows show the training distributions in feature space (delta distributions), while the solid lines are the actual densities learned by the model. Both size and location are very sharp distributions, while proportion of red color is a smoother Gaussian. Also note that the size distribution (**middle**) is skewed, similar to numerosity.

### 4.1.3 Size and Location

We also use the pie dataset in Figure 4 to explore size and location. Similar to color, we fix all training images to have either a given size, or a given location, while varying the other features randomly. The results are shown in Figure 5(middle and right). Interestingly we observe that the size distribution is skewed, and the model has more tendency to produce larger objects, while the location distribution is fairly symmetric.

### 4.2 Convolution Effect and Prototype Enhancement Effect

Now that we have probed the algorithm's behavior when $p(\mathbf{z})$ is unimodal, we investigate its behavior when $p(\mathbf{z})$ is multi-modal. We find that in feature space the output distribution can be very well characterized by convolving the input distribution with the the learning algorithm's output on each individual mode (impulse response), if the input modes are far from each other (similar to a linear filter in signal processing). However we find that this no longer holds when the impulses are close to each other, where we observe that the model generates a unimodal and more concentrated distribution than convolution would predict. We call this effect prototype enhancement in analogy with the prototype enhancement effect in cognitive psychology [32, 10].

**Experimental Settings**    For these experiments we use the color proportion feature of the pie dataset in Figure 4. We train the model with two bimodal distributions, one with 30% or 40% red (two close modes), and the other with 30% or 90% red (two distant modes). We also explore several other choices of feature/modes in the appendix, and they show essentially identical patterns.

**Results and Discussion** The results are illustrated in Figure 6. When the training distribution is sufficiently close (top row), the modes "snap" together and the mean of the two modes is assigned high probability. That is, objects with 35% red are the most likely to be generated, even though they never appeared in the training set. When the modes are far from each other, convolution predicts the model's behavior very well. Again, these results are consistent for GAN/VAE and different architectures/hyper-parameters (Appendix A).

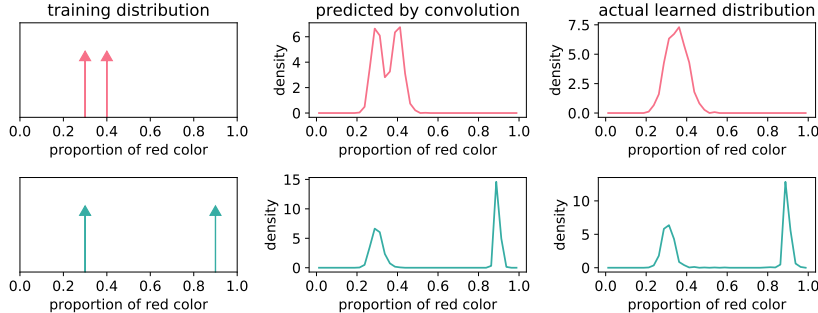

Figure 6: Illustration of the convolution and prototype enhancement effects. **Left:** The training distribution $p(z)$ consists of either two modes close together (top) or far from each other (bottom). **Middle:** The predicted response by convolving the unimodal response with the training distribution. **Right:** The actual $q(\mathbf{z})$ the learning algorithm produces. When the two modes are far from each other, $q(\mathbf{z})$ is very accurately modeled by convolution, while for modes close together, the resulting distribution is more concentrated around the mean.

Similar observations have also been made in psychological experiments. For example, for a set of similar examples the participant is more likely to identify the "average" (which they haven't seen) as belonging to the set of examples compared to actual training examples (which they have seen) [10].

## 4.3 Independence of Features

In this section we show that each of the features we consider can be analyzed independently of the other. We find that the generalization behavior along a particular feature dimension is fairly stable as we modify the distribution in other dimensions. As a result, we can decompose the analysis across dimensions. In fact, Section 4.1.1 already presented some evidence of independence where we showed that learned distribution on numerosity is similar for both dots and CLEVR.

**Experimental Setting** For these experiments we use the pie dataset in Figure 4. We study all three features: proportion of red color $\mathbf{z}_{\text{red}}$, size $\mathbf{z}_{\text{size}}$, and location $\mathbf{z}_{\text{loc}}$, and show that the learning algorithm's response on each is independent of the other features. For each feature, we select three fixed values (0.3, 0.4, 0.9 for proportion of red color, 0.5 0.55 0.8 for size, and -0.05 0.0 0.2 for location). For each fixed value of the feature under study, the other features can take 1-50 random values. For example, when studying if generalization on proportion of red color $\mathbf{z}_{\text{red}}$ is independent of other features, the training distribution $p(\mathbf{z}_{\text{red}}, \mathbf{z}_{\text{size}}, \mathbf{z}_{\text{loc}})$ is chosen such that the marginal on $p(\mathbf{z}_{\text{red}})$ is uniform on $\{0.3, 0.4, 0.9\}$. If $\mathbf{z}_{\text{red}}$ is independent on of the other features, the learned distribution $q(\mathbf{z}_{\text{red}})$ should only depend on this marginal $p(\mathbf{z}_{\text{red}})$ but not $p(\mathbf{z}_{\text{size}}, \mathbf{z}_{\text{loc}}|\mathbf{z}_{\text{red}})$. To explore different options for $p(\mathbf{z}_{\text{size}}, \mathbf{z}_{\text{loc}}|\mathbf{z}_{\text{red}})$ we select 1-50 random values as the support of this conditional distribution. This covers a very wide range of interactions between $\mathbf{z}_{\text{red}}$ and the other two features from strongly correlated (1 value) to very weakly correlated (50 values).

**Results and Discussion** The learned distribution for each feature as the other features vary is shown in Figure 7. We find that the learned distribution for each feature is fairly independent of the other dimensions. The only notable change is that there is a slight increase in variance if the other dimensions are more random. Interestingly, as the variance increases, modes that did not demonstrate prototype enhancement are starting to merge, verifying our previous conclusions. These results are consistent for GAN/VAE and also CNN/FC networks (Appendix A).

## 5 Characterizing Generalization on Multiple Features

In this section we are interested in the joint distribution over multiple features. As we discussed in Section 3, the combinations a dataset $\mathcal{D}$ covers can be exponentially small compared to all possible

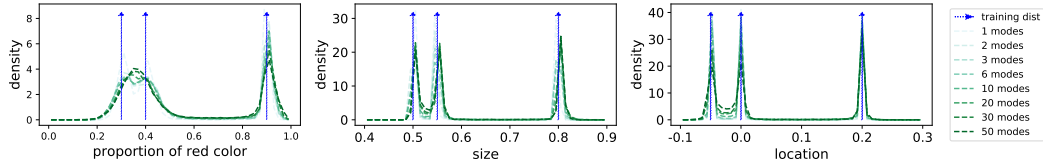

Figure 7: **From left to right:** Results on proportion of red color, size, and location. For all three features, the learned distribution (green lines) is relative independent of how many modes the other "nuisance" features can take (1-50). The only notable difference is that the learned distribution has higher variance if the other "nuisance" features are more random (more modes). As variance increases, some modes may display the tendency of merging together (prototype enhancement).

combinations in the underlying population distribution $p$. Therefore we explore when a learning algorithm trained on a few combinations can generalize to novel ones. We find that if the training distribution only contains a small number of combinations (e.g., 10-20) (in a feature space $\mathcal{Z}$) the learned distribution memorizes them almost exactly. However as there are more combinations in the training set, the model starts to generate novel ones. We find this behavior to be very consistent across different settings.

**Experimental Setup**   We use three different datasets:

**1)** Pie Dataset: We use the pie dataset as shown in Figure 4. There are four features: size (5 possible values), x location (9 possible values), y location (9 values), proportion of red color (5 values). There are a total of approximately 2000 possible combinations, and we randomly select from 10 to 400 combinations as $p(\mathbf{z})$ to generate our training set.

**2)** Three MNIST: We use images that contain three MNIST digits. For each training example we first randomly sample a three digit number between 000 to 999, then for each digit, we sample a random MNIST belonging to that class (random style). There are 1000 combinations, and we randomly select from 10 to 160 of them to generate our training set. Example samples are shown in Appendix A.

**3)** Two object CLEVR: We use the CLEVR dataset where each object has two properties: its geometric shape and its color. On this dataset, we select one shape that takes only a quarter of the possible colors, and one color that is assigned to only a quarter of the possible shapes.

**Evaluating Precision-Recall**   For pie and MNIST datasets, we use precision-recall to compare the support (evaluation detail in Appendix B) of $p(\mathbf{z})$ and $q(\mathbf{z})$ (Figure 8 Left). Recall is defined as the proportion of combinations in the support of $p(\mathbf{z})$ that is also in the support of $q(\mathbf{z})$. A perfect recall means that all combinations in $p(\mathbf{z})$ appears in the learned distribution $q(\mathbf{z})$. Precision is defined as the proportion of combinations in the support of $q(\mathbf{z})$ that is also in the support of $p(\mathbf{z})$. A perfect precision means that the learned distribution only generates combinations in the training set.

The precision-recall between support of $p(\mathbf{z})$ and $q(\mathbf{z})$ is shown in Figure 8. It can be observed that both GAN and VAE achieve very high recall on both datasets (pie and MNIST). This means that there is no mode missing, and essentially all combinations in the training set are captured by the learned distribution. However as the number of combinations increases the precision decreases, implying that the model generates a significant number of novel combinations. This means that if the desired generalization behavior is to produce novel combinations, one does not need a large number of existing combinations, and approximately 100 is sufficient. However this can also be problematic if one wants to memorize a large number of combinations. For example, some objects may only take certain colors (e.g., swans are not black), and in natural language some words can only be followed by certain other words. How to control the memorization/generalization preference for different tasks is an important research question.

We show in Figure 25 (Appendix) the results are independent of the size of the network. We obtain almost identical IoU curves from small networks with 3M parameters to large networks with 24M. In addition, the results are independent of the size of the dataset, and no difference was observed with only half or twice as many training examples. In this task, low precision appears to be inherent for GAN/VAEs and cannot be remedied by increased network capacity or more data.

**Visualizing Generalization**   For the CLEVR dataset, we precisely characterize how $q(\mathbf{z})$ differs from $p(\mathbf{z})$. We use a training set where a shape only takes a quarter of the possible colors and observe its possible generalization to other colors. The results are shown in Figure 9. First, we find that similar

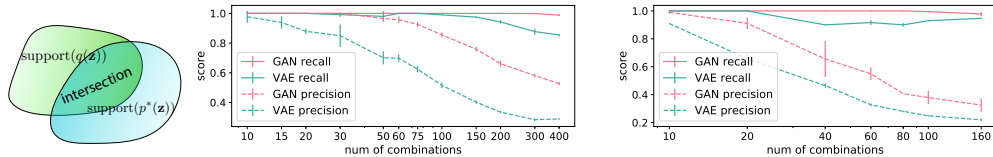

Figure 8: Precision and recall for GAN/VAE as a function of the number of training combinations. **Left:** recall measures intersection/support($p(\mathbf{z})$), while precision measures intersection/support($q(\mathbf{z})$). **Middle:** precision and recall for pie dataset. **Right:** precision and recall for three MNIST dataset. Both models achieve very high recall (the support of $q(\mathbf{z})$ contains the support of $p(\mathbf{z})$), but precision decreases as the number of combinations increases. Lower precision means that the learned distribution generates combinations that did not appear in the training set.

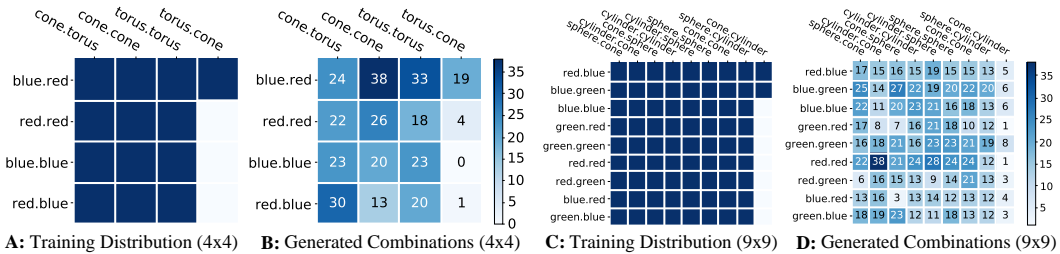

| A: Training Distribution (4x4) | B: Generated Combinations (4x4) | C: Training Distribution (9x9) | D: Generated Combinations (9x9) |

Figure 9: Generalization result on CLEVR where we have both shape and color as features. **A:** Training distribution is uniform, except if the shape is torus-cone, the torus must be blue and the cone must be red in 4x4. **B:** The marginal is preserved in the learned distribution (e.g. approx. the same proportion of torus-cone in both $p(\mathbf{z})$ and $q(\mathbf{z})$), and a small number of novel colors are occasionally generated. **C:** For 9x9, if the shape is cone-cylinder, it must take either red-blue or blue-green. **D:** The shape takes on all colors almost evenly, indicating clear generalization to all combinations.

to the pie and MNIST experiments, the generalization behavior critically depends on the number of existing combinations. If there are few combinations (e.g. 16), then the learning algorithm will generate very few, if not none, of the combinations that did not appear. If there are more combinations (e.g. 81), then this shape does generalize to all colors. What is interesting to note is that the marginal is approximately preserved, so each shape is generated approximately as many times as it appeared in the training set. When a rare shape (a shape that appeared in few colors) generalizes to other colors, it "shares" the probability mass with colors that did not appear. Similar experiments with rare colors are shown in Appendix A.

# 6 Conclusion and Future Work

In this paper we proposed an approach to study generative modeling algorithms (of images) using carefully designed training sets. By observing the learned distribution, we gained new insights into the generalization behavior of these models. We found distinct generalization patterns for each individual feature, some of which have similarities with cognitive experiments on humans and primates. We also found general principles for single feature generalization (convolution effect, prototype enhancement, and independence). For multiple features we explored when the model generates novel combinations, and found strong dependence on the number of existing combinations in the training set. In addition we visualized the learned distribution and found that novel combinations are generated while preserving the marginal on each individual feature.

We hope that the framework and methodology we propose will stimulate further investigation into the empirical behavior of generative modeling algorithms, as several questions are still open. The first question is what is the key ingredient that leads to the behaviors we have observed, since we explored two types of models (GAN/VAE), both of which have two very different architectures, training objectives, and hyper-parameter choices (Appendix A). Another important direction is to study the interaction between a larger group of features. We have been able to characterize the model's generalization behavior on low dimensional feature spaces, while generative modeling algorithms should be able to model thousands of features to capture distributions in the natural world. How to organize and partition such a large number of features remains an open question.

**Acknowledgements**

This research was supported by Intel, TRI, NSF (#1651565, #1522054, #1733686), and ONR.

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
