[Supplementary Material]

# A Extended Figures for Different Objectives and Architectures

To ensure that our conclusions are not sensitive to hyper-parameter choices or architecture choices, we also explore a very different architecture/hyper-parameter for each model. The following show the properties of the models we investigate. We found the models to have qualitatively similar behavior.

|  | WGAN | WGAN-FC | VAE | VAE-FC |
|---|---|---|---|---|
| convolutional | yes | no | yes | no |
| objective | WGAN-GP | WGAN-GP | ELBO | $\beta$-VAE, $\beta = 3$ |
| generator batch norm | no | yes | yes | yes |
| initial learning rate | 1e-4 | 1e-4 | 1e-4 | 5e-3 |
| latent noise type | Gaussian | Bernoulli | Gaussian | Gaussian |

Table 1: WGAN and VAE are the models we used throughout the main paper. We additional explore two very different architecture/hyper-parameter (WGAN-FC, VAE-FC) to ensure that our conclusions are not sensitive to these modeling choices.

## A.1 Extended Figures for WGAN

This is the primary setting we use in all of our experiments in the main body. The architecture and training detail is almost identical to [26].

Figure 10: Extended Plot of WGAN to Figure 6

## A.2 Extended Figures for VAE

This is the primary VAE setting we use in all VAE experiments in the main body. Additional plots not in the main body are supplied here.

Figure 11: Corresponding Figure for VAE to Figure 3

Figure 12: Corresponding Figure for VAE to Figure 5

Figure 13: Corresponding Figure for VAE to Figure 6

Figure 14: Corresponding Figure for VAE to Figure 7

## A.3 Extended Figures for WGAN-FC

In addition we explore WGAN-GP very different architecture/hyper-parameter. We use a fully connected architecture and Bernoulli latent noise. We show that the choice of these hyper-parameters do not affect our results. We still use WGAN-GP objective because we observe that DCGANs suffer from severe mode missing and is not suitable for our experiments

Figure 15: Corresponding Figure for WGAN-FC to Figure 5

Figure 16: Corresponding Figure for WGAN-FC to Figure 6

Figure 17: Corresponding Figure for WGAN-FC to Figure 7

Figure 18: Corresponding Figure for WGAN-FC to Figure 8

## A.4 Extended Figures for VAE-FC

We explore an additional setting for VAE. We use fully connected encoder and decoders, a larger learning rate, and increase the coefficient of $D_{\mathrm{KL}}(q(z|x)\|p(z))$ term to 3. [33].

Figure 19: Corresponding Figure for VAE-FC to Figure 5

Figure 20: Corresponding Figure for WGAN-FC to Figure 6

Figure 21: Corresponding Figure for WGAN-FC to Figure 7

Figure 22: Corresponding Figure for VAE-FC to Figure 8

## A.5 Extended Figures for CLEVR

We show the generalization result for color on the Clevr dataset in Figure 23. Similar to the shape experiment, we explore the generalization behavior when there are 4x4 (left two figures) and 9x9 possible configurations (right two figures). In each setting, the left figure represents the training set, and the right shows the frequency of generated features.

Figure 23: Generalization result for the color feature on Clevr dataset, the setting is identical to Figure 9.

## A.6 Samples for Three MNIST Dataset

Figure 24 shows four samples from the Three MNIST Dataset in Section 5.

Figure 24: Four sample images in the Three MNIST Dataset

## A.7 Invariance to Architecture Size

In Figure 25 we show that memorization is independent of the size of the deep network.

# B Experiment Details

## B.1 Datasets

**Dots:** For the dots dataset in Figure 2 we generate generate $k$ dots with color and location sampled uniformly at random. To obtain non-overlap images we reject the sample if any two dots overlap.

Figure 25: Precision-Recall curve for WGAN for different architecture sizes for the experiment in Figure 8 middle. We obtain almost identical results from small networks with 3 millions parameters to large networks with 24 million.

**CLEVR:** For the CLEVR dataset in Figure 2 we generate $k$ CELVR objects using the public implementation in [29]. For each object we use uniformly random shape (cylinder, sphere or cube), color, location, and size. We reject the image if any two objects overlap by more than 10%.

**Pie:** For the pie dataset in FIgure 4 we generate each image by starting with a pie with given percentage of red color and three other uniformly randomly selected non-red colors (red component is 0). Each color initially occupy a fixed slice in the circle. Then we swap a slice with random angle (size) and random location with another slice with identical angle (size) at another random location. We repeat the swapping process four times. This allows for more variation in the dataset.

## B.2 Evaluation

To evaluate the size of a circle in Figure 4, we compute the non-background area and then use the area to compute the corresponding radius. This is effective because the training images (and generated samples) contain a single colored circle with white background, so we can easily identify the background through its white color.

To evaluate the location of a circle in Figure 4 we compute the average location of the non-background pixels, and to evaluate the proportion of red color, we use the number of non-background pixels with larger $R$ component than $G$ or $B$ on the RGB space.

To evaluate the support of $q(\mathbf{z})$ in Section 5, to avoid the effect of possible error and noise when computing $\phi$, a feature combination $\mathbf{z}^{(i)} \in \mathcal{Z}$ is evaluated to be within the support of $\text{support}(q(\mathbf{z}))$ if $q$ assigns to $\mathbf{z}^{(i)}$ at least 10% the uniform probability $p^*(\mathbf{z})$ assigns to each of $\mathbf{z}^{(i)}$ in the training set.

To evaluate the MNIST digit in Section 5, we use a CNN trained with shift/rotation augmented training. The CNN produces high accuracy predictions (>95%) on generated samples based on human judgment of 500 samples.