[Reviews · NeurIPS 2018]

Reviewer 1



After reading author responses and discussing with other reviewers, I have decided to raise my score. I think the authors did a good job in their response to the points I raised. However, I still think that their should be more emphasis in the paper on the significance of the observations made in the paper which was not clear to me at first. -------------------------------------------------- This paper presents an empirical analysis of some probabilistic generative models, including GAN (specifically WGAN-GP) and VAE. The study relies on probative experiments using synthetic image datasets (e.g. CLEVR, colored dots, pie shapes with various color proportions) in which observations can be explained by few, independent factors or features (e.g. shape, color, numerosity). The paper aims at quantifying generalization in these models through their generated samples under various training scenarios, such as: - Training data has a single mode (e.g. only two CLEVR objects in the scene) - Training data has two distinct modes (e.g. pie shapes with either 30% or 90% red proportions) - Training data has two close modes (e.g. pie shapes with either 30% or 40% red proportions) - Other experiments investigate s multiple features vary in training data, such as shape and color in two CLEVR objects. Results show that the models exhibit some general behavior patterns. Notably, these patterns seem to be consistent across different training objectives (GAN or VAE), architecture and hyper-parameters. Strengths • The paper is clearly written • The paper investigates many interesting questions with well designed experiments Weaknesses • A fundamental weakness of this paper is the lack of a clear "hypothesis" for explaining observations. While it's interesting to note convolution effect or prototype enhancement, it is not clear how they can be explained or how they could be used for understanding these models and possibly improving them. • While it is interesting to draw similarities between generalization behaviour of probabilistic generative models and results in cognitive psychology, it is not clear what we could conclude from such resemblance. Can we claim we have an "intelligent" model? • The statement that the observed generalization behavior was consistent across different modelling objectives and architectures is unsettling. First, it seems to be based only on 4 experiments (as shown in supplementary materials -- Table 1). What does this tell us about these studied models? Does it extend to different models (e.g. autoregressive ones) or other variants of GANs or VAEs? In conclusion, I believe this paper presents some interesting observations but lacks a proposal for exploiting these observations, either for improving generative models or evaluating them.

Reviewer 2



Update: I thank the authors for their thorough response. I'm now further convinced of the significance of their work, especially with the addition of experiments varying network capacity and dataset size. I've increased my score accordingly. ------ The paper analyzes the behavior of generative models of images on synthetic datasets where latent factors like object count, size, and color are known. The authors observe that for each latent factor, the model distribution generally has greater spread than the original dataset. Another main result is that models tend not to learn complicated joint distributions over several latent factors when they are highly multimodal, instead partially reverting to modeling the factors independently. The findings are consistent across VAEs and GANs with fully-connected and convolutional architectures. Despite tremendous recent advances in generative modeling, our best models remain poorly understood and have potentially-problematic behaviors which we're not aware of. This paper makes a solid contribution towards characterizing and understanding generative model behavior. Further, the empirical methods introduced (generative modeling on synthetic image datasets with known latent factors) can be useful to future work which seeks to evaluate generative models. The experiments are thorough, well-described, and generally convincing. Some experiments are similar to previous work in generative model evaluation, though the ones in this paper are generally more thorough: - Many of the experiments are similar in spirit to the "visual hyperplane task" in [1], though that work was in a different context. The precision-recall experiments are similar to ones from [2] in the context of model evaluation. - The consistency of the results across VAE/GAN might be slightly overstated; Figure 8 shows that there is some gap in the extent of the effect, though not in the overall trend. One potential empirical question is how much of the spread observed in e.g. Fig 5 results from estimation error in the measurements rather than modeling error? A note to clarify this would improve the overall argument. All of the algorithms studied are asymptotically consistent; in the limit of infinite data and model capacity, they will exactly recover the data distribution. This paper is about generalization and inductive biases with finite data and model capacity; given that, an important question is precisely how large these networks are, and how big the datasets are (it's only mentioned in passing that they're trained on hundreds of thousands of distinct examples''). At the least, the authors should include these experimental details. The paper would be stronger with an experiment on how the observed effects vary with dataset size and model capacity -- do the effects drop off at all as we scale up, or should we expect them to hold for the largest models we can feasibly train? The experiments are generally clearly described and the claims follow directly from the experiments. The authors do a good job of connecting their work to cognitive science literature without overstating the connection. The clarity of the presentation could be improved somewhat by explaining the indended meaning of generalization'' -- in this paper it means something slightly different from the typical sense (i.e. the model hasn't overfit to the training set). Overall I recommend acceptance. [1] "Parametric Adversarial Divergences are Good Task Losses for Generative Modeling", Huang et al. [2] "Are GANs Created Equal? A Large-Scale Study", Lucic et al.

Reviewer 3



* Summary: This paper provides an empirical study of the generalization properties of deep implicit generative models. Previous works have mostly focused on the mathematical generalization properties of such networks [Arora et al., ICML17], while previous empirical studies lacked the rigorous methodology of this paper. This paper introduces a methodology inspired by neuroscience to probe such models, by using toy datasets with specified high-level features. For instance, for a dataset of images made of small dots, the number of dots and the colors of the dots in each image are such features, which can be easily assessed. The paper investigates the differences between the distribution of such features in the dataset and their distribution in the generated dataset, across different models (VAE/GAN), architectures (fully connected / convolutional) and datasets. The patterns identified by the authors are the following: - Single features are generalized in a sort of convolutional way: the distribution "impulse response" of the algorithm to an input dirac distribution (parametrized by a real value, e.g. radius of circles in an image or proportion of red) behaves akin to a linear filter. When there are multiple diracs, the convolution effects still holds, with an enhancement effects for intermediate values when the diracs are close. This behaviour is close to previous neuroscientific findings. - Features are independent: modifying many other features does not change the distribution reponse of the algorithms when studying a single feature. - When multiple combinations of features are part of the dataset, the distribution responses keep the combinations in the original distribution (high recall), but generalize to new combinations when the number of combinations grows (precision decreasing). * Opinion: This is a paper of very good quality. Writing is very clear. I think that the value of this paper does not solely lie in its empirical findings - some of which are surprising and may spark debate - but also in the experimental methodology it proposes to assess the properties of generative models. This methodology would be a very significant contribution to help understand generalization for deep neural networks, provided it is reproducible in a proper fashion. * Questions : - Do the authors plan to release the experimental code along with the toy dataset (l.146)? At least, knowing the training scheme of all networks would be very helpful. - A weak point of the paper is the lack of discussion of the sizes of the datasets with respect to the sizes of the models. How many layers are used, how many channels/dimensions per layer? It is not sure that these values are adequate measures of the capacity of networks, but reporting them would help anyway. - The findings in sec 5 concerning the generalization growing with the number of combinations in the original distribution is very interesting. Is using absolute values (l. 229-257) for such combinations really relevant? Did the authors observe the same thresholds for varying architectures' sizes? - In Figures 9B and 9D, there are significant variations in the core set of generated combinations: for instance, in D, green.red/cone.sphere has 7, while red.red/cylinder.cone has 38. Yet the input distribution seems to be evenly distributed among all combinations. What could explain such a discrepancy? Is it due to a small sample size due to the fact that the shapes were evaluated by human judgement (and there were only 3 judges, cf l.118)? In this case, are the values in the last column really significant? Is it consistent across all architectures and algorithms, or are there differences? - I find very surprising that all results are consistent with both fully-connected networks and convolutional ones and VAE/GAN. There may be hidden biases in the methodology or the datasets in this case. What about other generative algorithms, such as PixelCNN?